# Ovarian Telomerase and Female Fertility

**DOI:** 10.3390/biomedicines9070842

**Published:** 2021-07-20

**Authors:** Simon Toupance, Anne-Julie Fattet, Simon N. Thornton, Athanase Benetos, Jean-Louis Guéant, Isabelle Koscinski

**Affiliations:** 1DCAC, Université de Lorraine, Inserm, 54000 Nancy, France; s.toupance@chru-nancy.fr (S.T.); simon.thornton@univ-lorraine.fr (S.N.T.); a.benetos@chru-nancy.fr (A.B.); 2Laboratory of Biology of Reproduction, University Hospital of Nancy, 54000 Nancy, France; annejulie1208@hotmail.fr; 3Université de Lorraine, CHRU Nancy, Pôle “Maladies du Vieillissement, Gérontologie et Soins Palliatifs”, 54000 Nancy, France; 4Inserm U1256, NGERE, Université de Lorraine, 54000 Nancy, France; jean-louis.gueant@univ-lorraine.fr

**Keywords:** telomerase, hTERT, female fertility

## Abstract

Women’s fertility is characterized both quantitatively and qualitatively mainly by the pool of ovarian follicles. Monthly, gonadotropins cause an intense multiplication of granulosa cells surrounding the oocyte. This step of follicular development requires a high proliferation ability for these cells. Telomere length plays a crucial role in the mitotic index of human cells. Hence, disrupting telomere homeostasis could directly affect women’s fertility. Strongly expressed in ovaries, telomerase is the most effective factor to limit telomeric attrition and preserve ovarian reserve. Considering these facts, two situations of infertility could be correlated with the length of telomeres and ovarian telomerase activity: PolyCystic Ovary Syndrome (PCOS), which is associated with a high density of small antral follicles, and Premature Ovarian Failure (POF), which is associated with a premature decrease in ovarian reserve. Several authors have studied this topic, expecting to find long telomeres and strong telomerase activity in PCOS and short telomeres and low telomerase activity in POF patients. Although the results of these studies are contradictory, telomere length and the ovarian telomerase impact in women’s fertility disorders appear obvious. In this context, our research perspectives aimed to explore the stimulation of ovarian telomerase to limit the decrease in the follicular pool while avoiding an increase in cancer risk.

## 1. Introduction

Female fertility is strongly linked to the number of ovarian follicles and their quality. The pool of ovarian primordial follicles is established during prenatal life and results from an intensive proliferation of ovogonia and somatic follicular cells that surround them. Follicular atresia begins even before birth and ends with menopause. The quality of the follicles is linked to their capacity to ovulate, to produce a fertilizable oocyte that leads to an evolutionary embryo, and finally to a healthy child. Folliculogenesis implies an intense proliferation of granulosa cells during follicular maturation. Overall, the quantity and quality of follicles depend greatly on the capacity of follicular cells to proliferate; any disruption of this mechanism will result in a shortened reproductive life span and ovulation disturbance. Telomeres, the nucleoprotein structures that cap chromosome extremities, play a crucial role in the proliferative potential of cells. Telomere length shortens with each cell division and triggers cell senescence when a certain threshold is reached. Therefore, telomere attrition is considered a primary hallmark of aging [1]. This review attempts to highlight the relation between telomere length/telomerase activity in ovarian cells physiologically and in two situations of infertility, PCOS and POF.

## 2. The Telomerase

Telomerase is a ribonucleoprotein complex that can offset telomere attrition by adding telomeric repeats to telomeres [2]. This enzyme is composed of two subunits, a catalytic unit, the telomerase reverse transcriptase (TERT), and a long non-coding RNA unit containing the template for telomere sequence, the telomerase RNA component (TERC) [3] (Figure 1).

In humans, telomerase is active in embryonic tissue, germinal cells [4], and stem cells [5], whereas most somatic cells lack telomerase activity [6] except for rare cell types like activated lymphocytes [7]. Telomerase is found reactivated in ~90% of cancers [6,8,9,10] (Figure 2). TERC is constitutively expressed in cells, whereas TERT is detected only in telomerase-positive cells [11,12]. Telomerase holoenzyme can extend telomeric sequences alone in vitro [13], but in vivo, accessory proteins are required for recruitment and binding of telomerase onto telomeres during the cell cycle [14]. Dyskerin, NOP10 (nucleolar protein 10), GAR1 (a small nucleolar ribonucleoprotein), and NHP2 (non-histone chromosome protein 2) stabilize the TERC subunit [15,16] (Figure 1), and the molecular chaperones HSP90 (heat shock protein 90) and P23 are required for TERT folding [17]. TCAB1 (telomerase Cajal body protein 1) accumulates telomerase in the Cajal body [18], a non-membrane nuclear organelle implicated in telomerase biogenesis and trafficking to telomeres [19]. In telomerase-positive cells, throughout the S-phase [20,21,22], the telomerase complex is recruited to telomeric DNA and adds approximately 60 nucleotides of telomeric repeats to the single-strand telomere overhang [22]. Short telomeres have a higher probability of being extended by telomerase than long ones [23], this being potentially explained by the inhibition of telomerase activity by the components of the shelterin complex associated with telomeres [24].

Telomerase also displays telomere-unrelated functions known as non-canonical telomerase roles [25]. Indeed, gene expression levels change in hTERT-immortalized cells, suggesting that telomerase may have an impact on a whole spectrum of biological activities [26]. Telomerase was shown to regulate NF-κB signaling pathways resulting in the stimulation of IL-6 (Interleukin-6), IL-8, and TNFα (tumor necrosis factor-α) [27]. It was also shown that TERT could act as a transcriptional modulator of the Wnt/β-catenin signaling pathway [28] and stimulate β-catenin-mediated epithelial-mesenchymal transition [29]. Moreover, TERT was shown to regulate vascular endothelial growth factor (VEGF) expression through binding to the transcription factor SP1 (Specific Protein 1) [30,31]. Nuclear TERT can also upregulate the expression of DNA methyltransferases through interaction with SP1 [32,33] and, through this way, silence the expression of some genes, especially tumor suppressors.

Under oxidative stress conditions, TERT can move out from the nucleus and localize to mitochondria [34,35]. In mitochondria, TERT can bind to mitochondrial DNA (mtDNA) and display a protective effect against reactive oxygen species, protecting cells against oxidative stress-induced damage [35,36,37]. TERT can also sensitize cells to oxidative stress by increasing levels of mtDNA damage, which may lead to apoptotic cell death [38]. This discrepancy in the mitochondrial role of TERT on oxidative stress can be explained by different levels of TERT expression. Mitochondrial TERT also exerts control on mitophagy by decreasing the processing of PINK1 (phosphatase and tensin homolog-induced putative kinase 1) [39]. TERC can also be imported into mitochondria [40], where it is converted to TERC-53 and exported back to the cytoplasm. TERC-53 acts as a signaling molecule involved in cell senescence [41].

## 3. Human Folliculogenesis, the Importance of Proliferative Capacity

Ovaries contain female gametes in special histologic structures named follicles, where the gamete (oocyte) is surrounded by somatic cells (follicular cells). Folliculogenesis comprises several steps summarized here and illustrated by Figure 3, with a special focus on granulosa cell proliferation.

### 3.1. Establishment of a Pool of Primordial Follicles at the Fetal Time

After their migration into the genital ridges, primordial germ cells undergo many mitoses from the 6th week of embryo development, with a maximum mitosis rate in the 3rd and 4th month, leading to the formation of primordial germ cell cysts. From the end of the 2nd month, variable numbers of ovogonia enter meiosis and block at the diplotene stage of the first meiosis division [42]. At the same time, these germ cell cysts break apart into oocytes that become surrounded by somatic pre-granulosa cells to form primordial follicles whose number is maximal at the 5th month of pregnancy in humans [42]. All these steps are enabled by the high proliferation capacity of ovogonia and follicular cells [43] and may be limited by short telomeres, restricting the size of the follicular pool [44].

### 3.2. Follicle Growth

From puberty until menopause, follicular development starts when follicles leave the pool of resting primordial follicles to enter the growth phase. From there, the early growing follicle will become dominant, undergoing a developmental process, including a dramatic course of cellular proliferation and differentiation [45]. This occurs similarly in all mammals: for example, bovine follicular cells of the primordial follicle divide 21 times until the stage of an antral follicle [46].

The next step of follicle maturation has also been well explored in bovine and implies a differentiation of granulosa cells into two different pools: granulosa cells building the wall of the antrum and granulosa cells forming the cumulus oophorus around the oocyte. In the antral wall, the proliferation rate is greater in the middle and antral layers than in the basal layer, where it seems similar, as observed in the cumulus oophorus [47].

This development of primordial into secondary follicles and subsequent maturation of the dominant follicle implies a very high proliferation capacity for granulosa cells [48]. The mitotic index of the granulosa cells continues to increase with follicular diameter before reaching a peak in the small antral follicles [45,49]; the granulosa cell number reaches to the hundreds of thousands before ovulation [50] (Figure 3). 

### 3.3. Telomerase and Follicle Growth

In this context, the telomere length of ovarian cells has crucial importance [48]. As the most efficient factor limiting telomere attrition, the telomerase presents, unsurprisingly, rather high activity in the ovaries, especially in granulosa cells, which may enable their numerous replications. This high telomerase activity was highlighted many years ago in cows [47] and recalls other multipotent stem cell characteristics of granulosa cells [51,52]; that is, they divide without the need of anchorage [53] and can be differentiated into varied cell types [54].

The activity pattern of the telomerase has been studied through the different types of follicles in ovaries and with aging, especially in large mammals (mainly in bovines and pigs) [55,56].

Heterogeneity among all follicles has been found; the highest levels of telomerase activity are found in the smaller, rapidly growing preantral follicles [49,55], followed by a decrease in activity with the maturation of the follicle [48]. This suggests that the high proliferative activity of granulosa cells could be partially linked to telomerase activity [57].

Interestingly, the study of telomerase in bovine antral follicles highlighted heterogeneity within the granulosa, with a pattern similar to the proliferation index. More importantly, the expression of the catalytic subunit of TERT appears conserved in the middle and next to the antrum layers (layers with a high proliferation rate), whereas it seems to be downregulated in basal layers and in the cumulus oophorus, where only residual telomerase RNA would persist [47,58], explaining a moderate proliferation rate during the last stage of follicle maturation. In pigs, comparable proliferation and TERT activity have been observed in granulosa cells of the wall of the antrum (expression of TERT only in middle and antral layers, and not in basal layers); however, TERT has also been detected in cumulus cells at a higher level than in bovine [56,59].

After ovulation, residual granulosa cells undergo differentiation to large luteal cells and enter apoptosis. Not surprisingly, they present a low telomerase activity. This evolution of granulosa cells, which have achieved their differentiation and are morphologically distinct from proliferating granulosa cells in early-stage follicles, can explain the decrease in telomerase activity, as reported in some studies [60].

### 3.4. Regulation of Telomerase Activity in Ovaries

#### 3.4.1. Effect of Age

Interestingly, aging significantly impacts ovarian telomerase activity in bovine [55], leading to a decrease in granulosa cells’ proliferative activity. In mice, ovarian TERT expression was found to decrease with aging [61]. In humans, ovarian TERT also decreases with aging and is associated with telomere attrition [62].

#### 3.4.2. Estradiol and Telomerase Activity

The relation between sex hormone and telomere length has been considered for some time, based on observations of longer telomeres in women than in men [63]. The sex-specific discrepancy of the dynamics of telomere length erosion throughout life [64] was first hypothesized, suggesting a protective impact of female sex hormones [65]. However, longitudinal studies are controversial, showing either higher or lower rates of telomere length (TL) attrition in women compared to same-aged men [66,67]. Recently, it was even suggested that the sex-related gap in TL could be established at birth since gender differences in TL are found in newborns [68].

However, these longitudinal data originate from leukocyte telomere length (LTL) analysis, and tissue-specific differences in TL dynamics can exist. During folliculogenesis, in ovarian aging, as well as during neoplastic development, the pattern of induced telomerase activity is dependent on cell type, time, and location [56].

A direct impact of estrogens has been suggested since the region of the *TERT* promoter presents an estrogen response element [67]. An indirect impact via the activation of *MYC*, which is a direct target of ESR, has been suggested by studies in granulosa of rats and mice [69,70]. Activation of the proto-oncogene *MYC* seems especially involved in ovarian carcinogenesis [57], like *BCL2* and *TERT*.

In addition, high levels of estrogens can act indirectly on TL through a reduction in reactive oxygen species [71]. Furthermore, epidermal growth factor (EGF), which stimulates telomerase activity, has been found in abundance in small follicles [59], suggesting its role in the regulation of the telomerase activity and telomere maintenance, particularly in small, rapidly growing follicles.

The estradiol impact on telomerase and telomere length dynamics has been explored by animal experiments: in rats, the administration of estrogens prevents the inhibition of telomerase, suggesting that telomerase withdrawal plays an integral role in granulosa cell apoptosis and follicular atresia induced by estrogen withdrawal [72]. On the contrary, estrogen deficiency in mice results in an inhibition of telomerase within ovaries in a tissue-specific manner. This leads to the shortening of telomeres and compromised proliferation of the follicular granulosa cells. These effects are offset by estrogen replacement therapy [73]. Interestingly, in return, telomerase could exert control on estrogen expression through the non-canonical pathway. A recent study showed that *in vitro* overexpression of TERT in granulosa cells enhanced the expression of steroidogenesis genes and increased the level of estrogen in pre-ovulatory luteinized granulosa cells [74].

Considering the importance of estrogen in the regulation of telomere length, especially through the regulation of telomerase activity, two physio-pathological situations can be studied from a telomerase point of view: the syndrome of polycystic ovaries (PCOS) relative to a very high ovarian follicular density, and high estrogen levels and premature ovarian failure (POF) relative to a very poor ovarian follicular density and poor estrogen levels. Based on the elements exposed above, long telomeres with high telomerase activity in PCOS patients and short telomeres with decreased telomerase activity in POF patients would be expected to be found.

## 4. Polycystic Ovary Syndrome (PCOS)

The syndrome of polycystic ovaries is the most common endocrine disorder causing female infertility. Among women of reproductive age, its prevalence is reported to be 6–8% in developing countries. PCOS is a multifactorial disorder, and symptoms include hyperandrogenemia, oligo-anovulation, and a polycystic morphology of the ovaries. In addition, oocyte quality is frequently impaired, and, finally, PCOS constitutes real ovulatory infertility. Furthermore, some metabolic disorders such as metabolic syndrome, type 2 diabetes, and cardiovascular disease are regularly associated with PCOS [75] due to a high incidence of insulin resistance (IR), which participates in the pathophysiology of this syndrome. The link between IR and sex hormones has been studied for over 20 years. IR is promoted more by high estrogen levels than high androgen levels, and hyperinsulinism seems to promote hyperandrogenism. Indeed, insulin can stimulate ovarian sex hormone secretion directly by linking to the ovarian insulin receptor and IGF-I receptor, and indirectly mainly by reducing the hepatic production of IGFBP-1 (the major circulating IGF-1 protein) [76,77]. PCOS is also associated with a proinflammatory state [78] since oxidative stress is not compensated by antioxidative factors [79]. The main symptoms and long-term conditions of PCOS are provided in Table 1.

PCOS pathophysiology is quite complex, and precise etiology remains to be elucidated. It results from environmental and genetic factors with a high degree of heritability. No single factor can fully explain the spectrum of abnormalities in PCOS; some studies reported that it might be related to longer telomere length in granulosa cells [75]. It has also been suggested that the persistent oxidative stress observed in PCOS may influence telomere homeostasis. Therefore, high inflammatory factors, especially reactive oxygen species, may increase telomeric attrition [80] and impair telomerase activity [81]. Telomere shortening may be another mechanism associated with the pathogenesis of PCOS and its comorbidities. In response to gonadotrophin treatment, PCOS patients frequently develop many follicles, but these follicles often contain oocytes of small size and with a low rate of maturation (metaphase II). Moreover, metaphase II oocytes from PCOS patients frequently present an impaired competence. Since the interaction between oocyte and surrounding granulosa cells is crucial for the maturation and acquisition of oocyte competence, oocyte abnormalities could result from an impaired interaction between oocyte and granulosa cells and/or an insufficient proliferation of granulosa cells. Insofar as telomere length and telomerase activity play a major role in folliculogenesis, some authors propose that telomere disruption and limited activity of telomerase might be involved in the PCOS syndrome [82].

Studies on the relation between TL and PCOS are contradictory. Many have focused on the peripheral blood leukocytes of PCOS patients and reported shorter LTL [78,83]. However, tissue-specific TL dynamics can exist, especially in ovaries that express telomerase activity.

Recently, Pedroso et al. [82] measured telomere length and telomerase activity in the ovarian cells of women with PCOS in comparison with women with regular menstrual cycles (control), who underwent in vitro fertilization with intracytoplasmic sperm injection (IVF/ICSI). They found no difference in the telomere length of cumulus cells from PCOS and non-PCOS women. However, telomerase activity in cumulus cells was higher in the PCOS group. In the immature oocytes (germinative vesicle and metaphase I), telomere length and telomerase activity did not differ between the two groups. Interestingly, telomere length in leukocytes was shorter in PCOS patients than in controls. The authors proposed that telomere length in cumulus cells from PCOS patients might be maintained thanks to a higher telomerase activity [82]. Moreover, the authors reported a positive correlation between androgen levels and telomerase activity in cumulus cells and concluded that androgens stimulated telomerase [82]. The mechanism by which androgens stimulate telomerase activity could involve aromatization and be linked to Erα, as proposed in hematopoietic cells [84]. The role of the high production of estrogen, characteristic of PCOS, is also very probable. Thus, androgens and estrogens contribute to telomere maintenance by increasing telomerase expression and activity.

Using a similar study design, Li et al. [85] found shorter granulosa cell TL but similar telomerase activity levels in PCOS patients. These observations suggest that the impact of other factors such as insulin resistance or oxidative stress on TL is important. However, the authors showed that patients with lower TA and shorter TL exhibited a longer duration of infertility and earlier onset of infertility symptoms [85]. This suggests that a reduction in TA in granulosa cells might be associated with ovarian function decline [57].

In contrast, Wei et al. [75] demonstrated significantly longer GTL in PCOS patients. This finding reinforced the possible impact of PCOS high hormone levels on telomere homeostasis [75]. Furthermore, as described above, higher telomerase activity is observed in the smallest preantral follicles, which are six times more abundant in PCOS ovaries than in other patients [47,72,86].

To conclude, the results of the different studies are controversial, and they are summarized in Table 2. Considering the variability of phenotypes associated with PCOS, many factors are implicated in the physiopathology of this syndrome. Although these studies demonstrate evidence of telomeric disruption in PCOS, the hyperandrogenism present in these women may compensate for the deleterious effect of the inflammatory state and metabolic disorders related to PCOS. To identify and clarify the implication of telomeric homeostasis in PCOS, further studies are required, with flawless methodology conducted on larger populations and using the most suitable method of TL measure.

## 5. Premature Ovarian Failure

Premature ovarian failure (POF) is a clinical syndrome defined by loss of ovarian function before the age of 40 years. Its prevalence is about 1% [87]. Symptoms include menstrual disturbance (oligo or amenorrhea) with raised gonadotrophins and decreased estradiol. As a diagnosis criterion, the European Society of Human Reproduction and Embryology (ESHRE) working group has suggested an oligo-amenorrhea of at least 4 months and FSH >25 IU/L twice, 4 weeks apart [88]. Currently, spontaneous pregnancy rates in women with idiopathic POF are low (between 5 and 10%), and there is no treatment to improve the chances of spontaneous conception [89]. The long-term consequences of POF are the same as menopause and include osteoporosis, cognitive dysfunction, urogenital symptoms, such as vaginal dryness, increased risk of cardiovascular and autoimmune disease, as well as all-cause mortality [90]. The symptoms and long-term consequences of POF are summarized in Table 3.

Short GTL and/or low GTA could limit the mitosis ability of granulosa cells and, therefore, impair oocyte competence, resulting in infertility since granulosa cells must divide intensely during follicular maturation [91,92,93].

Several studies have focused on the potential correlation between telomere length and/or telomerase activity in granulosa cells in women with POF. Some authors have hypothesized the existence of a possible link between LTL and female fertility since TL are synchronized in somatic tissues at birth and that strong correlations subsist later in life [94]. The confirmation of their hypothesis would lead to a POF predictive marker, more easily available than GTL. Unfortunately, the results are contradictory [95,96,97].

Looking for an explanation for these contradictory results, some authors have investigated telomerase activity in ovaries: In 2000, Kinugawa et al. [98] demonstrated that telomerase is active in ovaries and its activity normally decreased with age and in POF patients with follicle depletion. Butts et al. [60] and Xu et al. [99] found shorter telomeres and a lower TA in the granulosa cells of POF women than in healthy controls. These studies suggest that short telomeres could be associated with a fertility decrease and a shorter window of reproductive lifespan [60,99]. The hypothesis is that short telomeres could limit the proliferation capacity of granulosa cells and, in doing so, induce POF. Moreover, lower telomerase activity in granulosa cells would compensate less for telomeric attrition, leading to a limitation in the cell proliferation required for correct follicle maturation [48,60,98,100].

In conclusion, shorter telomeres and lower telomerase activity in granulosa cells seem to be associated with premature ovarian failure. The results are shown in Table 4.

These parameters could be an early and reliable marker of the decline in ovarian function. However, these results should be confirmed with larger studies with more subjects and the most suitable method [93]. The extrapolation of leukocyte telomere length does not seem possible, perhaps because initial telomere length and telomerase activity in granulosa cells can be impacted by some acquired factors such as in situ or general oxidative stress.

## 6. Discussion

As commonly agreed, telomere length/telomerase activity is a crucial factor in the proliferation capacity of cells in a tissue. Gonadal tissues present specificities in both sexes; in men, telomerase is constitutively activated throughout life in spermatogonia, whereas in women, TA partially compensating for telomere shortening declines with age in ovaries. “Ovarian aging” is, therefore, greatly reliant on TA in granulosa cells.

Based on this hypothesis, the ovarian reserve of follicles appears the result of a complex function in which the follicle number at a time “t” depends on the pool of follicles established at the fifth month of intrauterine life, with the cumulative number of follicles disappearing monthly until the time “t.” If the pool of primordial follicles established during the first months of pregnancy seems more reliable to ovogonial telomere length and ovogonial telomerase activity, the kinetics of disappearance of follicles after birth seems dependent on the telomere length/telomerase activity in granulosa cells. Mathematical models were used to predict infertility with aging according to TL and TA in granulosa cells. Predicted infertility was associated with either low levels of telomerase activity and short telomeres in GCs or with cumulative exposure to oxidative stress [101,102]. The conclusion of these models is the important role of sustained ovarian TA to preserve fertility in aging. Interestingly, it was shown that telomerase activity in luteinized granulosa cells was positively associated with pregnancy rate during *in vitro* fertilization (IVF) treatment [103]. Moreover, *TERT* SNPs are also associated with IVF outcomes [104]. In Dyskeratosis congenita (DKC), canonical human telomeropathy is characterized by low telomerase activity and short telomeres, and women display diminished ovarian reserves, as estimated by circulating levels of anti-Mullerian hormone [105] and impaired reproductive function [106]. These results are consistent with the hypothesis of granulosa telomerase activity as a potential mechanism to preserve fertility.

Although this theory appears confirmed by the studies of GTL and GTA in the case of POF, the results of studies provided in PCOS populations appear controversial. In PCOS, the initial ovarian reserve of primordial follicles is abnormally rich, but the proliferation capacity of granulosa cells is sometimes high, explaining the high rate of tweeness in PCOS populations and an increased risk of ovarian cancer [107]. In contrast, the proliferation capacity of granulose cells is probably sometimes diminished, explaining the high rate of small follicles with immature oocytes picked up after ovarian stimulation in the context of IVF. The variable expression of GTA could be explained by two major types of factors: on the one hand, high levels of estrogen and androgen, which increase telomerase activity via stimulation of the *TERT* promoter, and on the other hand, some factors such as oxidative stress, which are generated at a high level in PCOS patients, impair telomere length homeostasis or lead the cells to enter apoptosis via a non-canonical pathway, implying mitochondria dysfunction.

The impact of oxidative stress is probably major in cases of inflammatory disease of ovaries such as endometriosis with ovarian implants of endometrial tissue. In endometriosis, endometrial telomerase is abnormally activated because of the endogenous autonomous high estrogen secretion of endometrial cells linked to a constitutive abnormally stimulated aromatase [108]. This estrogen source stimulates the telomerase activity of endometrial cells, explaining the invasive and proliferative characteristics of these cells colonizing ovaries and targeting the inflammatory state at the origin of major oxidative stress. This oxidative stress strongly decreases TA in GCs [81] via NF-κB pathways, finally impairing oocyte quality. Li et al. proposed that intrafollicular TNF-α might also downregulate TA and TERT expression via the NF-κB pathway. Thus, special attention should be given to limit/avoid any situation leading to important oxidative stress in the ovaries.

As proposed by some authors, restoring ovarian activity could be obtained by inducing the apparition of new stem cells, inducing pluripotent stem cells (IPS) able to differentiate into oocytes, or stimulating the proliferation of existing ovarian cells [109] without increasing the risk of oncogenesis. Among the strategies to reactivate telomerase, the exposition of ovaries to four factors, identified by Yamanaka and Takahashi (Oct3/4, Sox2, c-Myc, and Klf4) [110], is often cited [109] since it appears sufficient to induce IPS. As presented above, c-Myc and Klf4 are known activators of TA [111,112] and could also reactivate dormant tissular stem cells. Animal experiments on telomerase activation have succeeded in increasing the longevity of mice without increasing the development of tumors [113]. Moreover, in aged telomerase-deficient mice, telomerase reactivation reversed tissue degeneration and allowed the restoration of animal fertility [114]. In humans, various regenerative assays have been provided. In some of them, platelet-rich plasma (PRP) has been used, restoring the proliferation capacity of dormant stem cells [115]. In some clinical trials, these therapeutics were implemented into routine clinical practice as a rejuvenating agent or to promote healing in dermatology, plastic surgery, dentistry, orthopedics, and, in a similar way, gynecology [116] in order to optimize IVF outcomes in POF patients. PRP contains many growth factors supporting the survey and proliferation of ovarian stem cells directly or indirectly through the activation of telomerase [117]. Nevertheless, by activating telomerase using either a genic therapy strategy or treatment, the risk of cancer seems a real threat [118].

Indeed, telomerase activity is found in all types of ovarian cancers [119,120]. Similar to the majority of telomerase-positive cancers, ovarian cancers display short TL and high TA [119]. Ovarian cancers originate from ovarian surface epithelial cells [121]. These cells are telomerase negative [120]. Telomerase activation is an early event in ovarian carcinogenesis [57], and the association of TERT expression with inactivation of p53 and activation of the Ras, Bcl2, and c-myc pathways are critical in oncogenic transformation [121]. Interestingly, these genes are estrogen-responsive genes [122,123]. Ovarian carcinogenesis could originate from estrogen paracrine activity. Estrogen from the granulosa cell compartment could reach the epithelial cells, inducing telomerase activation and oncogenic transformation of the ovarian epithelium [124]. The levels of telomerase activity correlate with the severity and recurrence of ovarian cancers [125]. Recently, effective telomerase-molecular targeted therapy has been tested as an approach for the treatment of ovarian cancer [126].

The presence of the *TERT* promoter in a dense CG-rich CpG island suggests a role for methylation in the regulation of TERT expression [127]. In cows, an age-dependent decrease in the global methylation stage correlates with a decrease in telomerase activity and a decrease in the proliferation capacity of granulosa cells [55].

Recently, some dietary recommendations have been proposed to improve telomerase activity [128] through their antioxidative role and impact on promotor methylation. However, investigations to understand the expression and methylation pattern of TERT in human ovaries [129] do not agree with animal studies; the methylation levels of the human *TERT* promoter do not seem to correlate with the level of TA. Therefore, all dietary recommendations, such as the Mediterranean diet, show that nutrient-promoting DNA methylations have probably more impact on TA through their antioxidative role than through their ability to improve ovarian TA. Similarly, the last decade has seen a great of studies analyzing the possible relation between the Mediterranean diet and TL. Most of the studies showed an association of the Mediterranean diet with longer telomeres [130,131]. However, these were almost exclusively cross-sectional and correlational studies, making causal interpretation problematic [132]. Longitudinal studies and randomized controlled trials are very limited, and their results are inconclusive [133,134]. Therefore, dietary recommendations may have an effect on ovarian aging without a direct effect on TA activity or TL, probably through oxidative stress modulation.

During adult life, in addition to their proliferative role in granulosa cells, TERT and TERC seem to globally protect ovarian mitochondria from oxidative stress [35,36,41], indirectly promoting mitochondrial steroidogenesis and, especially, estrogen synthesis. A non-proliferative role of both subunits of telomerase appears similar to a complementary amplification loop of its proliferative function since estrogen upregulates TERT expression [135]. Furthermore, a similar non-proliferative role of telomerase subunits may directly protect oocytes through their mitochondria, playing a role in oocyte competence, especially with aging [136]. Age-dependent mtDNA instability and the accumulation of mtDNA mutations in the oocyte result in the alteration of oocyte competence. Mitochondrial TERT may protect oocytes from these effects by binding to mtDNA.

## 7. Conclusions (Abstract Figure)

In the field of female fertility, telomerase presents a special interest since the capacity for cell proliferation is at the heart of embryonic development, particularly in fetal oocyte proliferation and during adult life proliferation of granulosa cells. Classical telomerase proliferative activity is upregulated mainly by estrogens by activation of the promoter and downregulated by different agents of oxidative stress. Moreover, through their interaction with mitochondria, telomerase subunits can more subtly modulate oocyte competence. The hypothesis in which telomerase activity correlates with ovarian reserves of follicles and global female fertility appears confirmed by pathology, especially premature ovarian failure, which is characterized by a poor ovarian reserve and low TA. However, PCOS, which is associated with a rich ovarian reserve, does not always match with long granulosa telomere length and high ovarian TA, probably because of the implication of downstream regulators such as oxidative stress.

Improving ovarian TA has been explored as a very promising rejuvenation technique. Unfortunately, no activator agent is without risk of activation telomerase in dormant cancer cells. At best, we can hope to limit the loss of natural fertility or infertility linked to a pathology by combating any factor jeopardizing the activity of telomerase. For this, hygiene of life and safe alimentation appear complementary to traditional strategies of fertility cryopreservation.

## Figures and Tables

**Figure 1 biomedicines-09-00842-f001:**
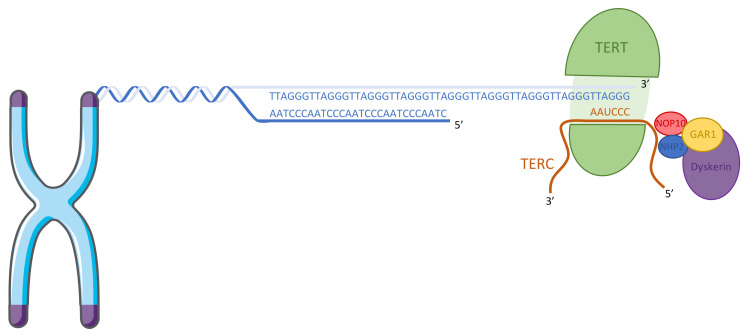
Structure of telomerase. The catalytic reverse transcriptase subunit TERT associate with the template RNA component TERC and is stabilized by the dyskerin complex (NOP10, NHP2, GAR1, and dyskerin).

**Figure 2 biomedicines-09-00842-f002:**
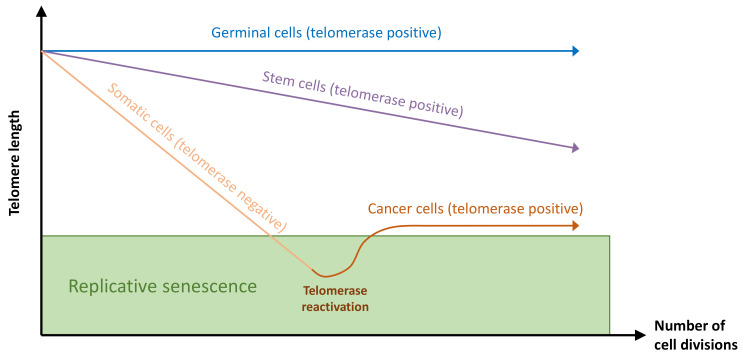
Telomere length dynamics in germinal, stem, somatic, and cancer cells according to the presence of telomerase activity. In germinal and stem cells, active telomerase can offset telomere attrition due to cell divisions. In somatic cells, telomerase is inactive, and telomere length is shortened until it reaches a threshold that triggers replicative senescence. Cancer cells can bypass cell senescence by the reactivation of telomerase.

**Figure 3 biomedicines-09-00842-f003:**
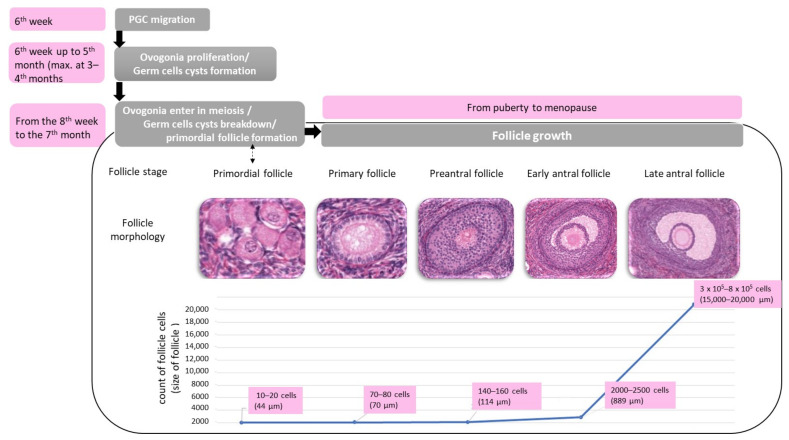
Human folliculogenesis and granulosa cells proliferation. The figure describes the different stages of follicle growth and the corresponding granulosa cell count and follicle size. Granulosa cells need to display a high proliferative potential to accommodate the tremendous proliferation needed during folliculogenesis.

**Table 1 biomedicines-09-00842-t001:** Diagnostic criteria according to AE-PCOS Society (2006) and long-term consequences [77].

	Clinic and Biologic Symptoms	Long-Term Clinical Consequences
Diagnostic criteria according to Androgen Excess-PCOS Society (2006)	Biochemical and clinical hyperandrogenism (hirsutism and acne)	Hirsutism
	Ovarian dysfunction (oligo-amenorrhea)	Infertility
	Polycystic ovary morphology	
	Insulin resistance/increased incidence of type 2 diabetes	Metabolic syndrome
	Increased incidence of cardiovascular disease	

**Table 2 biomedicines-09-00842-t002:** Studies measuring telomere length/telomerase activity in granulosa cells in PCOS.

Authors	Variable Measured/Method	Population No.	Population Age (Years)	Results
Li et al., 2017 [85]	GTL: PCR GTA: TRAP	(n = 65) PCOS (IVF) (n = 98) Non-PCOS (IVF)	30.3 ± 4.3 31.7 ± 3.8	GTL: shorter in PCOS GTA: no difference Earlier infertility symptoms when shorter TL and lower TA
Wei et al., 2017 [75]	GTL: PCR	(n = 75) PCOS (IVF) (n = 81) Non-PCOS (IVF)	28.36 ± 2.55 28.09 ± 2.26	GTL: longer in PCOS
Pedroso et al., 2020 [82]	GTL: PCR	(n = 43) PCOS (IVF) (n = 67) Non-PCOS (IVF)	33.7 ± 4.1	GTL: no difference GTA: higher in PCOS Similar TA and TL in GV and MI oocytes

GTL, granulosa telomere length; GTA, granulosa telomerase activity; IVF: *in vitro* fertilization; GV, germinative vesicle; MI, oocyte at the metaphase of the first meiosis division; TRAP, telomere repeat amplification protocol; PCR, polymerase chain reaction.

**Table 3 biomedicines-09-00842-t003:** POF symptoms and long-term conditions [93].

	Clinic and Biologic Symptoms	Long-Term Clinical Consequences
Diagnostic criteria according to ESHRE	Ovarian dysfunction (Oligo-amenorrhea for at least 4 months)	Infertility
	Clinical estrogen deficiency (climacteric syndrome) and sex hormonal dysfunction with: elevated FSH level >25 IU/L on two occasions >4 weeks apart	Urogenital symptoms (vaginal dryness, irritation and itching, sexual disorders) Cognitive dysfunction (memory and concentration problems, increased risk of dementia) Autoimmunity (increased risk of autoimmune disease) Bone alteration (osteopenia, osteoporosis, and increased risk of fracture) Increased all-cause mortality

**Table 4 biomedicines-09-00842-t004:** Studies associating telomere length and telomerase activity in granulosa cells/ovaries with POF.

Authors	Method	Population	Population Age (Years)	Results
Kinugawa et al., 2000 [98]	TRAP	(n = 5) POF (n = 20) Controls	31.4 <38	Ovarian TA: lower level in POF
Butts et al., 2009 [60]	PCR TRAP	(n = 12) POF (n = 42) Controls	30–37 23–37	GTL: shorter in POF GTA: absence of TA is more frequent in POF
Xu et al., 2017 [99]	PCR TRAP	(n = 120) POF (n = 90) Controls	32.95 ± 4.27 29.98 ± 4.28	GTL: shorter in POF GTA: lower level in POF

GTL, granulosa telomere length; GTA, granulosa telomerase activity; POF, premature ovarian failure; TRAP, telomere repeat amplification protocol; PCR, polymerase chain reaction.

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
