# Peer review of "Ovarian Telomerase and Female Fertility"

_biomedicines, 2021, doi:10.3390/biomedicines9070842_

Round 1

Reviewer 1 Report

The authors provide an overview of the links between ovarian telomerase and female fertility. The manuscript mainly focuses on the hypothesis that the telomere length has an impact on female conditions such as Polycystic Ovary Syndrome (PCOS) and Premature Ovarian Failure (POF). The hypothesis is interesting, but because of lack of systematic and accurate measurements of telomere length and contradictory studies, it is still questionable. I have several comments and suggestions I would like to see addressed prior to publication.

Major revisions

Different methods of measuring telomere length have their advantages and limitations.

Could you summarize the current methods used for measuring telomere length? FlowFISH seems to be the most accurate method (see DOI: 10.1371/journal.pone.0113747), but it wasn’t used in any of the studies that the authors cite in Table 2. 

Could you add the timeline for each step of follicullogenesis in Figure 3?

Page 12, lines 447-449: This sentence is confusing. How do telomerase subunits protect mitochondria from mtDNA instability?

Minor revisions

Page 1, line 21: “plays” and not “play”

Page 1, line 26: the abbreviation POF should be added just after Premature Ovarian Failure (used here for the first time)

Page 1, line 45: “chromosome” and not “chromosom”

Page 2, line 53: “sequencing” and not “sequenceing”

page 7, line 265: “leukocytes” and not “leucocytes”

Page 9, line 337: “leukocyte” and not “leucocyte”

Author Response

Answer to reviewer 1

Different methods of measuring telomere length have their advantages and limitations.

Could you summarize the current methods used for measuring telomere length? FlowFISH seems to be the most accurate method (see DOI: 10.1371/journal.pone.0113747), but it wasn’t used in any of the studies that the authors cite in Table 2. 

Answer: We totally agree with reviewer 1 that the methodology chosen to measure telomere length is a crucial point in the interpretation of the studies. This review focused on the role of telomerase, so we chose the studies according to this criterion (measurement of telomerase activity) and not according to the method used to measure telomere length. We recently made a systematic review of telomere length in POF specifically in which we addressed this point [1]. Briefly, there are several methods used to measure telomere length including the gold standard, Southern Blot, the high throughput qPCR, and fluorescent in situ hybridization-based methods qFISH and Flow-FISH. Each one has advantages and drawbacks [2–4]; Southern blot is the most accurate and gives results in absolute value (kb) but is labor intensive and expensive due to low throughput. qPCR use low amount of DNA and allow high throughput experiments, but it has a higher measurement error and gives results only as ratio. FISH methods allow distinction between cell populations but require fresh samples and are less precise. In research settings Southern Blot and qPCR are the most used techniques and one must privilege Southern blot for study with small effectives or longitudinal design [5–7]. In clinical settings such as in the field of telomere biology disorders where telomere length of patient is extremely short, FISH based methods can be privileged for diagnosis due to the good compromise between sensitivity, speed and automatization of the test [8]. We choose not to include this discussion in the review since we selected only studies that measure telomerase in ovaries and put aside all the studies focusing only on telomere length with different methodology. This review is not systematic concerning TL in ovaries and all the studies included are using the same method, qPCR. However, this method is not the most suitable for small effective studies and larger studies or study with better method choice are needed to confirm results as we point out in the text (lines 467-469 and 559, page8 and 9):

Page 8, 467-469: “To identify and clarify the implication of telomeric homeostasis in PCOS, further studies are required, with flawless methodology conducted on larger populations and using the most suitable method of TL measure.”

Page 9, 559: “and the most suitable method [Fattet et al., 2020]”.

  1. Fattet, A.-J.; Toupance, S.; Thornton, S.N.; Monnin, N.; Guéant, J.-L.; Benetos, A.; Koscinski, I. Telomere Length in Granulosa Cells and Leukocytes: A Potential Marker of Female Fertility? A Systematic Review of the Literature. J. Ovarian Res. 2020, 13, 96, doi:10.1186/s13048-020-00702-y.
  2. Aubert, G.; Hills, M.; Lansdorp, P.M. Telomere Length Measurement-Caveats and a Critical Assessment of the Available Technologies and Tools. Mutat. Res. 2012, 730, 59–67, doi:10.1016/j.mrfmmm.2011.04.003.
  3. Lai, T.-P.; Wright, W.E.; Shay, J.W. Comparison of Telomere Length Measurement Methods. Philos. Trans. R. Soc. Lond. B. Biol. Sci. 2018, 373, 20160451, doi:10.1098/rstb.2016.0451.
  4. Wang, Y.; Savage, S.A.; Alsaggaf, R.; Aubert, G.; Dagnall, C.L.; Spellman, S.R.; Lee, S.J.; Hicks, B.; Jones, K.; Katki, H.A.; et al. Telomere Length Calibration from QPCR Measurement: Limitations of Current Method. Cells 2018, 7, E183, doi:10.3390/cells7110183.
  5. Aviv, A.; Hunt, S.C.; Lin, J.; Cao, X.; Kimura, M.; Blackburn, E. Impartial Comparative Analysis of Measurement of Leukocyte Telomere Length/DNA Content by Southern Blots and QPCR. Nucleic Acids Res. 2011, 39, e134, doi:10.1093/nar/gkr634.
  6. Nettle, D.; Seeker, L.; Nussey, D.; Froy, H.; Bateson, M. Consequences of Measurement Error in QPCR Telomere Data: A Simulation Study. PloS One 2019, 14, e0216118, doi:10.1371/journal.pone.0216118.
  7. Nettle, D.; Gadalla, S.M.; Lai, T.-P.; Susser, E.; Bateson, M.; Aviv, A. Measurement of Telomere Length for Longitudinal Analysis: Implications of Assay Precision. Am. J. Epidemiol. 2021, 190, 1406–1413, doi:10.1093/aje/kwab025.
  8. Gutierrez-Rodrigues, F.; Santana-Lemos, B.A.; Scheucher, P.S.; Alves-Paiva, R.M.; Calado, R.T. Direct Comparison of Flow-FISH and QPCR as Diagnostic Tests for Telomere Length Measurement in Humans. PLOS ONE 2014, 9, e113747, doi:10.1371/journal.pone.0113747.

Could you add the timeline for each step of folliculogenesis in Figure 3?

Answer: Timeline indication has been added for each step of female gametogenesis in figure 3

Page 12, lines 447-449: This sentence is confusing. How do telomerase subunits protect mitochondria from mtDNA instability?

Answer: Mitochondrial TERT can bind to mtDNA and exert a protective function against oxidative stress-induced damage. This function can prevent from mtDNA instability. It is stated in lines 92-93 in the telomerase chapter. We reformulated this sentence (lines 109-112) and the one on lines 892-893 to clarify:

Page 3: Lines 109-112: “In mitochondria, TERT can bind to mitochondrial DNA (mtDNA) and display a protective effect against reactive oxygen species, protecting cells against oxidative stress-induced damage [35–36] or it can sensitize cells to oxidative stress, by increasing levels of mtDNA damage which may lead to apoptotic cell death [38].”

Page 12 Lines 905-906: “Mitochondrial TERT may protect oocyte from these effects by binding to mtDNA.”

Minor revisions

Page 1, line 20: “plays” and not “play”

Answer: “play” has been changed for “plays”

Page 1, line 25: the abbreviation POF should be added just after Premature Ovarian Failure (used here for the first time)

Answer: “(POF)” has been added just after Premature Ovarian Failure, since it was the first time it was written

Page 1, line 44: “chromosome” and not “chromosom”

Answer: “chromosom” has been changed for “chromosome”.

Page 2, line 61: “sequencing” and not “sequenceing”

Answer: “sequenceing” has been changed for “sequence”.

page 7, line 265 and Page 9, line 337: “leukocytes” and not “leucocytes”

Answer: “leucocytes” has been changed for “leukocytes” lines 427 (page 7) and 559 (page 9).

Reviewer 2 Report

This review deals with two hot topics in medicine and molecular biology: women’s fertility and possibility of the telomerase activity restoration. Authors collected and systematized a large amount of published data that could be of interest for the scientific and medical communities. Text is easy to read and understand.

The manuscript can be accepted for publication when following corrections are made:

  • Line 69

Please, specify in what S-phase moment telomeres are being extended by telomerase (at any moment of the S-phase or in the late S-phase, after replication).

  • - Paragraph from the line 80

TERT can regulate expression of several genes, but TERT itself is expressed only in the telomerase positive cells. Does it mean that TERT regulates expression only in the germinal cells?

  • Chapter 3, preamble, line 104

Please, check the Figure number. I think it should be Figure 3.

  • Figure 3

This figure should be clarified: coordinate axis designation is missing.

  • Line 171

On the Figure 2 we see that germinal cells are being constantly telomerase-positive during a number of cells divisions. At the same time at line 171 it is written that:

…In human, ovarian TERT decrease also with aging and is associated with telomere attrition…

Please, correct this discrepancy.

  • Line 254

Please, expand acronym LTL as it occur for the first time.

  • Line 256

This review tries to highlight the relation between telomere length/telomerase in ovarian cells and PCOS. 

I think that this sentence belongs to the Introduction part, rather than in the middle of the main text.

Author Response

Answer to reviewer 2

This review deals with two hot topics in medicine and molecular biology: women’s fertility and possibility of the telomerase activity restoration. Authors collected and systematized a large amount of published data that could be of interest for the scientific and medical communities. Text is easy to read and understand. The manuscript can be accepted for publication when following corrections are made:

- Line 69:

Please, specify in what S-phase moment telomeres are being extended by telomerase (at any moment of the S-phase or in the late S-phase, after replication).

Answer: In model organisms, yeast and ciliates, telomeres are elongated by telomerase activity during late S phase [1]. However, in human, it has been shown that telomerase activity on telomeres occurs during all S-phase [2–4].

  1. Shen, Z.-J.; Hsu, P.-H.; Su, Y.-T.; Yang, C.-W.; Kao, L.; Tseng, S.-F.; Tsai, M.-D.; Teng, S.-C. PP2A and Aurora Differentially Modify Cdc13 to Promote Telomerase Release from Telomeres at G2/M Phase. Nat. Commun. 2014, 5, 5312, doi:10.1038/ncomms6312.
  2. Tomlinson, R.L.; Ziegler, T.D.; Supakorndej, T.; Terns, R.M.; Terns, M.P. Cell Cycle-Regulated Trafficking of Human Telomerase to Telomeres. Mol. Biol. Cell 2006, 17, 955–965, doi:10.1091/mbc.E05-09-0903.
  3. Wu, P.; De Lange, T. Human Telomerase Caught in the Act. Cell 2009, 138, 432–434, doi:10.1016/j.cell.2009.07.018.
  4. Zhao, Y.; Sfeir, A.J.; Zou, Y.; Buseman, C.M.; Chow, T.T.; Shay, J.W.; Wright, W.E. Telomere Extension Occurs at Most Chromosome Ends and Is Uncoupled from Fill-in in Human Cancer Cells. Cell 2009, 138, 463–475, doi:10.1016/j.cell.2009.05.026.

The sentence has been clarified and reference 2-4 included as follows (lines 80-82):

“In human telomerase positive cells, throughout S-phase [20-22] the telomerase complex is recruited to telomeric DNA and adds approximately 60 nucleotides of telomeric repeats to the single-strand telomere overhang.”

- Paragraph from the line 80:

TERT can regulate expression of several genes, but TERT itself is expressed only in the telomerase positive cells. Does it mean that TERT regulates expression only in the germinal cells?

Answer: TERT can regulate gene expression in cells where it is expressed. These include male germinal cells (all life), female germinal cells (in the prenatal life), stem cells, some progenitor cells, granulosa cells, activated leukocytes and tumor cells.

In addition to a concomitant expression of TERT and DNMT3 during embryogenesis, development and cellular differentiation, TERT has been shown to regulate the expression of some genes, especially tumor suppressors by activating the promotor methylation of these tumor suppressors.

This aspect has been clarified page 3 from the line 108, as follow:” …and by this way, silent the expression of some genes, especially tumor suppressors.”.

- Chapter 3, preamble, line 104:

Please, check the Figure number. I think it should be Figure 3.

Answer: Line 123: “Figure 1” has been modified for “Figure 3”

- Figure 3:

This figure should be clarified: coordinate axis designation is missing.

Answer: The Y-axis has been specified as count of follicle cells. Follicular size is mentioned into brackets.

- Line 171:

On the Figure 2 we see that germinal cells are being constantly telomerase-positive during a number of cells divisions. At the same time at line 171 it is written that: “…In human, ovarian TERT decrease also with aging and is associated with telomere attrition…”. Please, correct this discrepancy.

This is not a discrepancy. TERT expression is decreasing with age in ovarian tissue. Ovarian tissue is not constituted only of germinal cells. One explanation could be the decrease of TERT expression in Granulosa cells.

- Line 254:

Please, expand acronym LTL as it occurs for the first time.

Answer: “Leukocyte Telomere Length” is mentioned before the first use of LTL at the line 273 page 5.

- Line 256:

“This review tries to highlight the relation between telomere length/telomerase in ovarian cells and PCOS”. I think that this sentence belongs to the Introduction part, rather than in the middle of the main text.

Answer: We removed this sentence from line 420 (page 7) and placed it, slightly modified at the end of the introduction part line 57-58 (page 2)

“This review tries to highlight the relation between telomere length/telomerase activity in ovarian cells in two situations of infertility, PCOS and POF.”

Round 2

Reviewer 1 Report

The authors have addressed my concerns.